# RoboHive: A Unified Framework for Robot Learning

**Vikash Kumar**[ικλγφ]*, **Rutav Shah**[δ], **Gaoyue Zhou**[γ], **Vincent Moens**[φ], **Vittorio Caggiano**[φ],
**Jay Vakil**[φ], **Abhishek Gupta**[λζι], **Aravind Rajeswaran**[ιλκζφ]

U.Washington[ι], UC Berkeley[λ], CMU[γ], UT Austin[δ], OpenAI[κ], GoogleAI[ζ], Meta-AI[φ]
https://sites.google.com/view/robohive⬀

## Abstract

We present RoboHive, a comprehensive software platform and ecosystem for research in the field of Robot Learning and Embodied Artificial Intelligence. Our platform encompasses a diverse range of pre-existing and novel environments, including dexterous manipulation with the Shadow Hand, whole-arm manipulation tasks with Franka and Fetch robots, quadruped locomotion, among others. Included environments are organized within and cover multiple domains such as hand manipulation, locomotion, multi-task, multi-agent, muscles, etc. In comparison to prior works, RoboHive offers a streamlined and unified task interface taking dependency on only a minimal set of well-maintained packages, features tasks with high physics fidelity and rich visual diversity, and supports common hardware drivers for real-world deployment. The unified interface of RoboHive offers a convenient and accessible abstraction for algorithmic research in imitation, reinforcement, multi-task, and hierarchical learning. Furthermore, RoboHive includes expert demonstrations and baseline results for most environments, providing a standard for benchmarking and comparisons. Details: https://sites.google.com/view/robohive⬀

## 1   Introduction

Recent years have witnessed unprecedented breakthroughs in Artificial Intelligence (AI), particularly in the domains of game playing [1, 2], protein folding [3], and language modeling [4]. Comparatively, the progress in robot learning has been slow. This slower pace can partially be attributed to Moravec's paradox [5], which posits that sensorimotor behaviors are intrinsically harder for AI agents than high-level cognitive tasks. Simultaneously, another crucial issue demands our attention: a significant hindrance also lies in the convoluted software frameworks for robot learning and the lack of universally recognized benchmarks. This increases the barrier for entry, limits rapid prototyping, and restricts the influx of ideas. Unlike fields such as computer vision or natural language processing, where benchmarks and datasets are standardized, the landscape of robotics remains more fragmented.

Addressing this gap, we introduce RoboHive, a cohesive ecosystem tailored for robot learning.

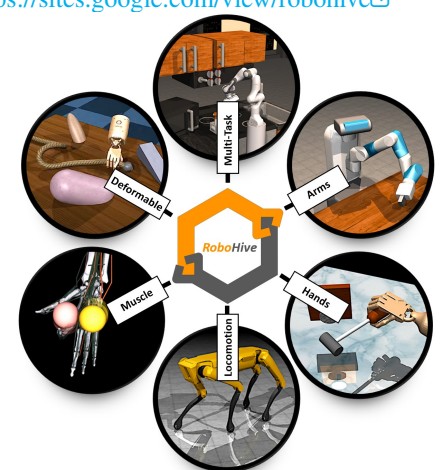

Figure 1: A subset of RoboHive's task-suites. [in clockwise order] **Multi-Task Suite**: Environments facilitating multiple tasks at once. **Arms Suite**: Diverse arms (w/ grippers) exposed to tabletop manipulation tasks. **Hand Suite**: Diverse hands exposed to tasks requiring dexterity. **Locomotion Suite**: Diverse collection of legged locomotion tasks. **Myo Suite**: Task collections with musculoskeletal agents. **Deformable Suite**: Tasks collection with deformable objects.

As a dual-function platform, RoboHive operates as a benchmarking suite and a research tool. It delivers a plethora of environments, precise task

---

*Correspondence to vikashplus@gmail.com

37th Conference on Neural Information Processing Systems (NeurIPS 2023) Track on Datasets and Benchmarks.

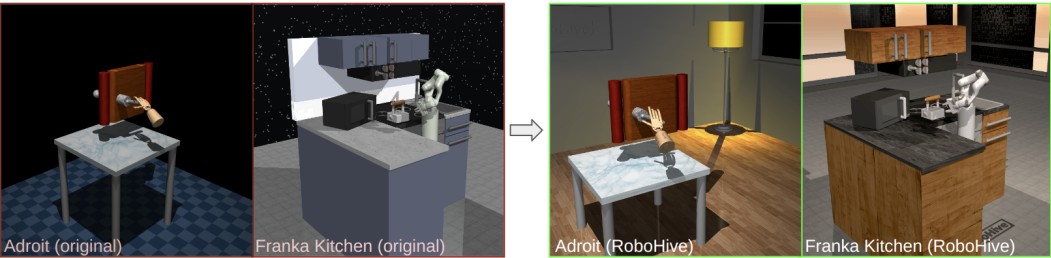

Figure 2: `RoboHive` presents environments with rich visual diversity to aid research in visual generalization and places high emphasis on physical realism to facilitate the real-world transfer.

definitions, and stringent evaluation criteria, supporting a diversity of learning paradigms like reinforcement, imitation, and transfer learning. This facilitates efficient exploration and prototyping for researchers. Additionally, `RoboHive` equips users with teleoperation capabilities and hardware integration, enabling a seamless transition between physical robots and their simulated counterparts. With `RoboHive`, our goal is to bridge the gap between the current state of robot learning and its potential for growth.

The **primary contribution** of our work is the development and open sourcing of the `RoboHive`, a unified framework for robot learning. The key features of `RoboHive` include:

1. **Environment Zoo:** `RoboHive` features an extensive and diverse collection of environments, covering a broad spectrum of research areas. These environments encompass a wide range of manipulation tasks involving both free and articulated objects, dexterous in-hand manipulation, locomotion with bipedal and quadrupedal robots, and even manipulation with musculoskeletal arm-hand models. Our simulated environments, powered by MuJoCo [6], provide fast physics simulation and are designed to ensure a high level of physical realism.

2. **Unified Robot Class and out-of-box hardware support:** `RoboHive` introduces a unified Robot-Class abstraction that seamlessly interfaces with both simulated and physical robots using sim-hooks and hardware-hooks, respectively. This unique capability enables researchers to effortlessly interface with robotic hardware, and port their results from simulation to reality by simply modifying a single flag.

3. **Teleoperation Support and Expert Dataset:** `RoboHive` offers teleoperation capabilities out-of-the-box through various modalities, including a keyboard, 3D space mouse, and virtual reality controllers. Collected using this feature, we are open-sourcing *RoboSet – one of the largest real-world manipulation datasets* collected via human-teleoperation covering 12 skills across multiple kitchen tasks. These teleoperation features and datasets are particularly valuable to researchers engaged in imitation learning, offline learning, and related fields.

4. **Physics Fidelity and Visual Diversity:** To expose the next research frontier in real-world robotics, `RoboHive` emphasizes tasks with high physics fidelity and rich visual diversity surpassing previous benchmarks. We incorporate complex assets, rich textures, and improved scene composition, aligning visuomotor control research with real-world visual complexities. Furthermore, `RoboHive` natively supports visual domain randomization and scene layout randomization in several environments, enhancing the versatility of visual perception, while also providing realistic and rich physical content.

5. **Metrics and Baselines:** `RoboHive` establishes clear and concise metrics for evaluating algorithm performance across all environments. The framework provides a user-friendly gym-like API [7] for seamless integration with learning algorithms, ensuring accessibility for a wide range of researchers and practitioners. Moreover, in collaboration with TorchRL [8] and mjRL [9], `RoboHive` includes comprehensive baseline results☑ for commonly studied algorithms within the research community, offering a benchmark for performance comparison and analysis.

## 2   Relationship to Prior Work

Benchmarks play a crucial role in guiding research progress. In the realm of robot learning, simulated state-based benchmarks like OpenAI Gym [7] and dm-control [10] have played a pivotal role in leading the development cycles of numerous algorithms and applications [11, 12, 13, 14, 15, 16].

However, with the advancements in the field, benchmarks also need to evolve to calibrate the difficulty and rigor of the problems (to avoid overfitting), as well as to expose the next research frontiers. For instance, the progression from MNIST towards CIFAR-10 in favor of ImageNet significantly influenced the evolution of deep learning for computer vision [17, 18, 19, 20, 21]. Similarly, RoboHive aims to provide an enhanced set of comprehensive and challenging benchmarks representative of the next frontiers of research in the current robot learning landscape.

Unlike other fields, like NLP and vision, where research questions are well-defined and often homogeneous, open questions in the field of Robot learning comprise of diverse and heterogeneous sets: such as (bipedal/quadrupedal) locomotion, (legged/wheeled, arial/terrestrial/aquatic) navigation, (on/off-road) driving, (rigid/deformable, in-hand/full-body) manipulation. Given this diversity, the field relies on a cluster of benchmarks each with unique strengths. Despite this, unification can facilitate the cross-pollination of ideas that can significantly catalyze progress.

Towards this goal, RoboHive offers a comprehensive ecosystem for robot learning, serving as both a diverse collection of benchmarks and a comprehensive research toolkit with out-of-the-box support for common robotic hardware drivers, teleoperation capabilities, one of the largest real-world robotics datasets, etc. to name a few.

In terms of benchmarking, RoboHive presents a diverse collection of environments and tasks under a unified framework that includes domains like dexterous and tabletop manipulation, legged locomotion, musculoskeletal agents, deformable objects, multi-task, multi-agent problem statements, as well as real-world benchmarks in locomotion and manipulation. This is in contrast to existing benchmark clusters like [7, 10] which primarily consist of state-based simple toy domains unrepresentative of realistic robotics challenges. RoboHive tasks are designed with attention to physical realism which provides an advantage over the [22, 23, 24, 25] cluster of benchmarks, which trades off physical accuracy for large-scale scenes and photorealism. While there are efforts that adhere to physical realism [26, 27, 28] they are often domain-specific. RoboHive greatly surpasses their task and visual diversity, and covers multiple domains. The work that bears the closest resemblance to ours is IsaacGym [29], which features photo-realistic rendering, and GPU-accelerated physics. However, these advantages come at the cost of physical realism and demand a high level of expertise for development. This can be particularly challenging for research investigations that frequently require customization and rapid prototyping.

Lastly, several environments in RoboHive have been adapted from prior works and are made available with the permission of the original authors. These environments were not originally designed and released as benchmarks [30, 31, 32, 33], but have emerged as such over time. Wide adoption without support led to various customization by the community leading to fragmentation and challenges when comparing results across different versions. In the spirit of the evolution of such well-proven benchmarks, RoboHive adopts these environments and presents them within a single framework. The refreshed versions of these environments feature rich visual diversity (Figure 2) to meet the growing needs for visual generalization in robot learning, clear evaluation metrics, and baselines to facilitate further adoption by the community. The associated datasets have also been reparsed, checked for reproducibility, and are being made available with RoboHive. The unification of these tasks under RoboHive not only brings them together under a single framework but also deprecates the unsupported dependencies making the entire framework be instantiated via `pip install robohive`.

## 3  Design Philosophy & Framework Overview

The primary goal of RoboHive is the development and open-sourcing of a unified framework to support research in robot learning. Alongside a wide collection of carefully designed tasks, RoboHive provides an ecosystem to bridge the gap between algorithmic advancements in our benchmarks and real-world results. An overview of the different components of RoboHive is outlined in Figure 3. We provide additional details on these components and the rationale behind their fundamental design principles in this section.

### 3.1  Abstract Robot Class for Simulation & Hardware Backends

At the core of RoboHive is an abstract robot class, depicted in Figure 4, which provides a unified abstraction for both simulator and hardware backends. The abstract robot class exposes unified APIs for various sensors and robots across both simulation and hardware, thereby abstracting low-level

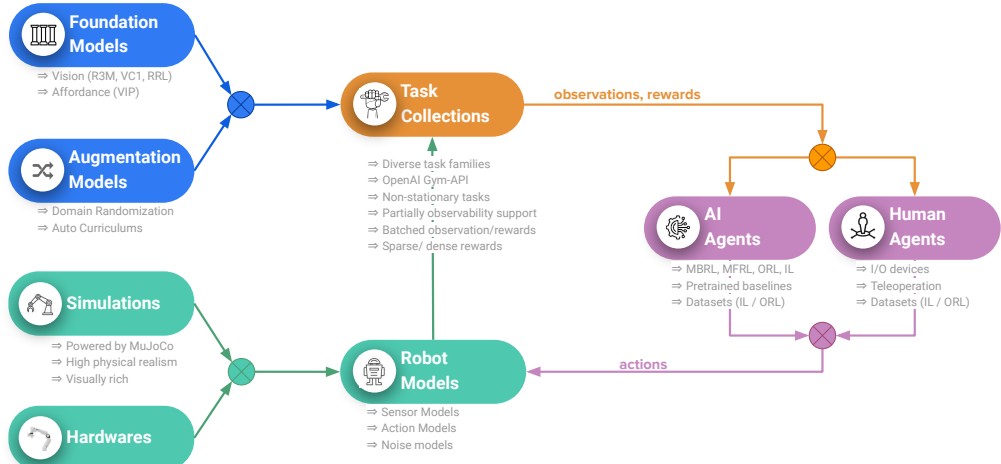

Figure 3: `RoboHive`'s versatility stems from its modular design. The four primary modules are – *task-collections* that contain multiple environment suites, *agents* take contain human as well as algorithmic actors, a unified *robot* that seamlessly bridges simulation and reality, and a *foundation/augmentation* module that supplements `RoboHive`capabilities with pre-trained models and sim2real paradigms.

implementations and enabling the switch between simulation and the real world by toggling just a single flag. For a live demo of this functionality, please see the linked video↗. Currently, we support a diverse collection of commonly studied robots across several morphologies as outlined below.

- **Hands**: Adroit [34], Shadow [33], Allegro [35], MPL [36], D'Hand [37], D'Manus [38]
- **Arms**: Franka [31], Sawyer [37], Fetch [39]
- **Quadrupeds**: Spot Mini [40], MIT-Cheetah [41], D'Kitty [32]
- **Bipeds**: Darwin [42], Atlas [43]
- **Musculo-skeletal**: MyoHand [44], MyoElbow [44]

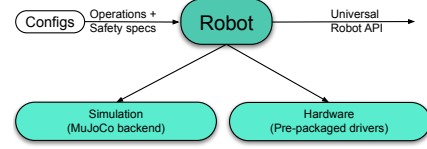

Figure 4: Overview of `RoboHive`'s robot class that presents a unified framework to seamlessly work between simulation and physical hardware

**Simulation**   Since the broader vision of `RoboHive` is to facilitate progress in real-world robot learning, all simulation models are designed with care toward alignment with real-world considerations. To ensure a high degree of physical realism, we build `RoboHive` on top of the now open-source MuJoCo physics engine [6]. Furthermore, `RoboHive` natively supports domain randomization, sensor noise, and delays, which can all be easily toggled through a base configuration file (Figure 4).

**Hardware**   `RoboHive` provides out-of-the box support for various types of robot hardware. It also exposes a base class for users to extend as per the hardware configurations they have access to. The *robot-hardware* class exposes robots in the real world via respective drivers. The *robot* class supports multiple control modalities such as position, velocity, and torque control, and can be easily customized using a configuration file. Hardware and sensor drivers and hooks for common robots are natively supported for easy deployment and transfer of results from simulation to the real world.

## 3.2   Environments

`RoboHive` exposes environments through the widely adopted OpenAI Gym API. All the environments are organized into suites, a visual depiction of which is provided in Figure 5. They span from whole arm and dexterous manipulation of free and articulated objects to locomotion agents on varied terrains, to control of detailed musculoskeletal models. These environments are useful across different algorithmic approaches such as visual imitation learning, model-free, and model-based RL.

For the **observation space**, `RoboHive` environments expose observations through abstract sensors for both simulation and the real world. Sensors can include low-dimension states (simulation-only), proprioception, and cameras (RGBD). We also natively support common transformations of the visual observations. This includes **visual domain randomization** [45] of textures and lighting conditions,

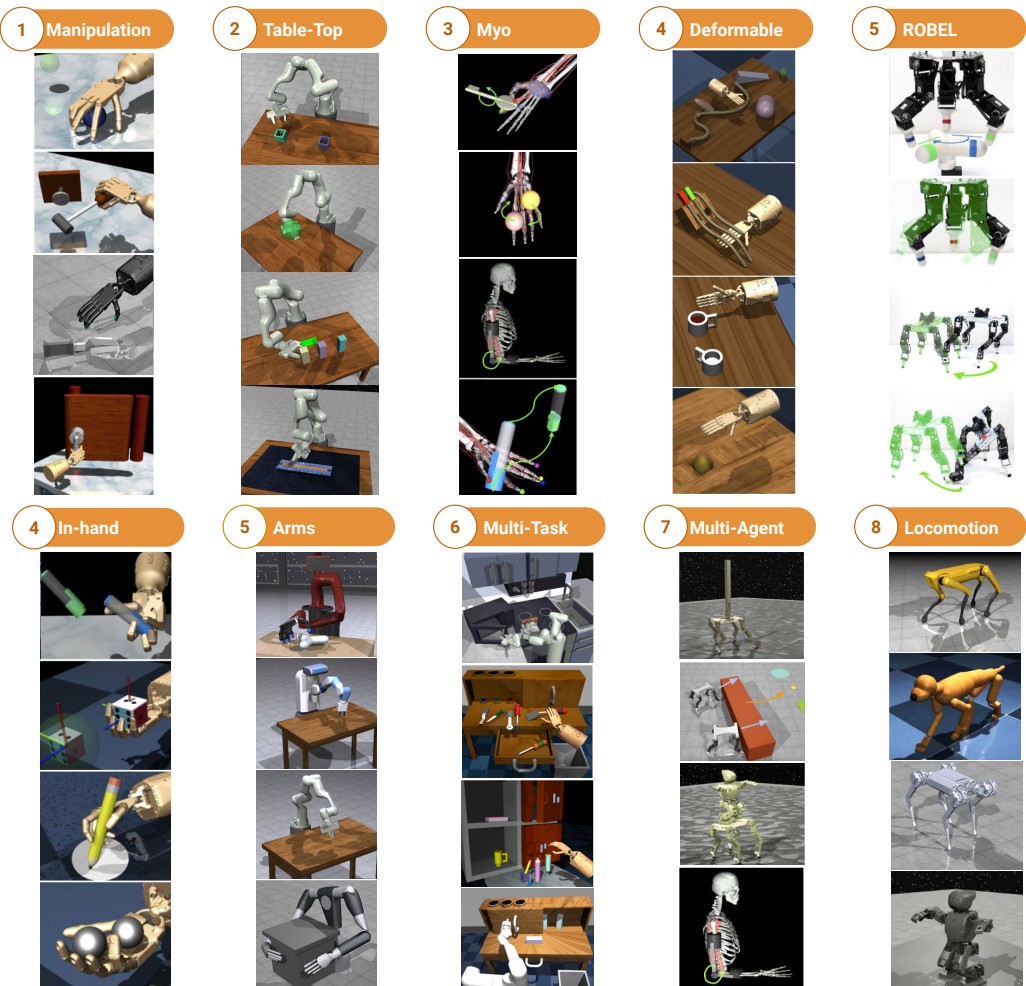

Figure 5: `RoboHive` Framework hosts a wide collection of environments of varying complexity (state/visual observations, fully/partially observable, sparse/dense rewards, etc) across multiple domains. Each column represents a task family within `RoboHive` and highlights a subset of its tasks.

as well as embedding the visual observation using **pre-trained visual models** [46, 47, 48, 49, 50]. During registration, `RoboHive` environments can be fully customized using a base configuration file, to equip the robot with various sensors, augmentations, and visual foundation models.

`RoboHive` also explicitly decouples **rewards and success criteria**. In RL, it is common to use rewards to judge a task's performance. This complicates relative comparisons as developing reward functions (either by hand or through learning) that can generate desired robot behaviors is itself a core component of the robot learning field. Thus, `RoboHive` ships with an independent *success criteria* per environment that can be used for evaluations, as opposed to reporting rewards. We provide basic dense reward functions for all tasks and facilitate users to improve or provide new reward functions. Finally, `RoboHive` also supports fully vectorized/batched rollouts in simulation for **fast environment sampling**, along with named dictionaries for sensors, observations, and reward/success metrics to enhance **interpretability** and to facilitate debugging.

**TaskSuite Overview:**    Next we provide a brief overview of various task suites present in `RoboHive`.

**Hand Manipulation Suite**    This task suite contains a collection of hands. The set of tasks is inspired by [51] and presents a collection where the arm and hand joints need to be coordinated to solve tasks. Tasks in this set typically have a high action space and face challenges in temporally co-ordaining small movements of the fingers with large movements of the arms.

**Table-Top Suite**    This task suite contains a collection of tasks involving arms and grippers interacting with small-scale objects. Tasks like stacking blocks, pouring, zipping, pushing, and pick-place are common in this task suite.

**Myo Suite**    MyoSuite contains a unique collection of tasks involving musculoskeletal systems. This collection is inspired by [44] and challenges policies with third-order actuation involving muscle dynamics and high dimensional actuation space of MyoHand. Tasks of varying difficulty levels are present – posing, pointing, key turn, baoding balls, etc.

**Deformable Manipulation Suite**    Our deformable suite contains a set of tasks involving flexible material . MuJoCo is primarily a rigid body simulator but has support for a few deformable objects such as ropes and soft bodies. The tasks in this suite involve interacting with ropes, granular medium, soft sponges, etc. These tasks need to be solved from visual inputs as the state space of the deformable objects is hard to specify.

**Robel Suite**    ROBEL suite was introduce in [32] and consists of well defined manipulation and locomotion tasks using *D'Claw* and *D'Kitty* low cost robots with parallel instantiation in real world.

**In-hand Manipulation Suite**    In hand manipulation suite is primarily inspired by the task set presented in [33]. Tasks in this set predominately involve in-hand reorientation of objects such as die, baoding balls, pencils, etc. Catastrophic failures due to the loss of control and thereby the object pose a primary challenge in solving this suite.

**Arm Manipulation Suite**    Arm manipulation suite is similar in spirit to the tabletop suite but requires tasks that require careful attention to the morphology of the arms. Tasks in this set involve interaction with large objects (such as appliances) where self-collisions and collision with objects are of concern.

**Multi-Task Suite**    With one of the primary motivations behind `RoboHive` being generalization, this suite is of key importance. This suite is organized as a collection of tasks where an agent can have shared experiences that can be generalized between tasks. This task suite builds off from the task set introduced in Franka-kitchen [31]) and table scene (as introduced in [52])

**Multi-Agent Suite**    To facilitate investigation in multi-agent interactive behaviors, we present a novel suite in `RoboHive`. Tasks in this suite build from tasks presented in [32], [53]. Tasks objective include interacting with passive objects, as well as other agents.

**Locomotion Suite**    `RoboHive` also features a wide collection of locomotion agents and a variety of locomotion tasks . Both agents and tasks are inter-changeable in this suite. Task objectives vary from simple reaching tasks to maneuvering around obstacles as well as object interaction.

**Classical Suite**    Finally, no task collection is complete without a set of simple-to-understand classical problems . While such problems are well represented in existing benchmarks [7, 10], `RoboHive` hosts a set that exposes unique challenges such as - non-holonomic control, over actuation, etc.

### 3.3    Agents

By exposing environments through the OpenAI Gym API, `RoboHive` makes it convenient to develop various agents/algorithms that can help the robots accomplish various tasks. In the realm of reinforcement learning, by instantiating an environment (through a base configuration file) with an observation space and reward as described in Section 3.2, along with a robot's native action space, any standard RL algorithm applicable to an MDP/POMDP setting can be used to train agents. Along with TorchRL [8] and mjRL [9], we provide an initial set of baseline implementations with policies effective in both simulations⚹ as well as the real world⚹. Code samples of these implementations along with baseline results for various environments can be directly accessed from https://sites.google.com/view/robohive⚹.

Besides online RL, robot learning encompasses other classes of algorithmic approaches to train agents, such as offline RL and imitation learning. These approaches however require a starter dataset from the environment with at least a few successful trajectories. To facilitate the collection of such datasets, `RoboHive` provides out-of-the-box **teleoperation support** for most environments through a variety of user interfaces like keyboard, 3D space mouse, and VR controllers. This teleoperation capability is shared across both simulation and the real world. Please see this short live demo⚹ that demonstrates the ease of teleoperation capabilities shipped with `RoboHive` in under 2 minutes, right from pip installing the package.

## 4    Capabilities and Utilities of the `RoboHive` Framework

We believe that `RoboHive` has several capabilities and components that are of broad use to various communities related to robot learning. We outline several of the salient capabilities and utilities of `RoboHive` in this section.

## 4.1 `RoboHive` as a repository of environments and agents

**Environments** `RoboHive` naturally provides a unified framework for environments and agents that can power research in robot learning. By providing a large diversity and complexity of robotics simulation assets and tasks, fast and physically realistic simulation models, and a modular framework that is easy-to-use and extend, `RoboHive` naturally plugs into the ecosystem meeting research needs in reinforcement learning and imitation learning settings. Given the need for more complex tasks and visually rich environments for studying visual generalization, and the huge space of users who rely on MuJoCo for their simulation needs [2], `RoboHive` provides the ecosystem necessary to both develop learning algorithms as well as deploy them on various robots both in simulation as well as real-world studies. By matching environments with real-world considerations and by providing both simulation and hardware backends and hooks, `RoboHive` can considerably reduce the barrier of entry to robot learning and enable researchers primarily interested in algorithm development to more easily deploy their ideas on robots. `RoboHive` provides this flexibility without sacrificing speed and accuracy. To demonstrate the performance of `RoboHive`, we provide the environment throughput for several task suites in Table 1 (See Table 3 for full details).

Table 1: This table illustrates the environment data collection performance on a single machine using 32 processes distributed across 8 A100 GPUs. The data collection process utilizes TorchRL's collector classes in an asynchronous manner, following a "first ready first served" approach. Each batch of data (500 steps/batch, 800K total) was collected on a separate process, employing a random policy, and transformed to a floating point tensor and resized to $[84 \times 84]$ pixels. The results presented in this figure are based on three different seeds, and the reported values indicate the average and standard deviation across seeds and environments. Multi-task suite rendering was achieved over 4 cameras with a resolution of $[256 \times 256]$ pixels, Arm and Hand suites had 3 cameras with a resolution of $[244 \times 244]$ pixels. The Myo suite does not support rendering over dedicated cameras and was excluded from this analysis. The decision to employ TorchRL as the backend for data collection is justified by the substantial superiority of these results compared to those achieved using single-process or multiprocessed solutions offered by most other frameworks, as reported in more details in [8].

| Task Family | Samples or Frames per second | |
|---|---|---|
| | State observations | Proprioception + Visual observations |
| Arm Manipulation | $7473 \pm 1237$ | $993 \pm 63$ |
| Hand manipulation | $8785 \pm 665$ | $1713 \pm 24$ |
| Multi-task | $4430 \pm 6$ | $346 \pm 0.4$ |
| Myo | $7366 \pm 3509$ | n/a |

**Agents** `RoboHive` supports two types of behavioral agents: (1) human agents (teleoperation), and (2) algorithmic agents (e.g. policies trained with reinforcement learning or imitation learning).

**Human Agents (teleoperation):** `RoboHive` natively supports various input devices or interfaces using which humans can interact with `RoboHive` environments. Currently, supported interfaces include keyboards, game controllers, 6 DoF 3D connection space mice, VR controllers, and the versatile CyberGlove. These interfaces can be used to control both simulated real robots to accomplish various tasks through a teleoperation setup, and can also serve to generate datasets that can power various offline RL and imitation learning algorithms.

**Algorithmic Agents:** In addition to human agents, `RoboHive` also boasts a collection of pre-trained policies (with state as well as visual inputs) from well-studied algorithm families. These policies have been trained through partnering with torchRL and mjRL, and can be readily downloaded and used for interacting with the environment, generating datasets for offline RL or IL, or as a baseline to compare the performance of new algorithms. Code samples for training these policies and the pre-trained policies can be accessed from the project website ⬈.

## 4.2 Sim and Real Counterparts

To easily study transfer between simulation and reality, `RoboHive` makes it seamless to toggle between simulation and reality. Effective transfer between simulation and reality requires bridging the reality gap. `RoboHive` does this by providing near-photorealistic rendering, accurate physics modeling, and attention to detail in real-world factors such as latency, actuator, sensing delays, etc. Figure 6 (video ⬈) showcases a task-policy deployed in simulations as well real-world via the sim2real bridge across two domains.

---

[2] multiple simulation engines now natively support importing MuJoCo models

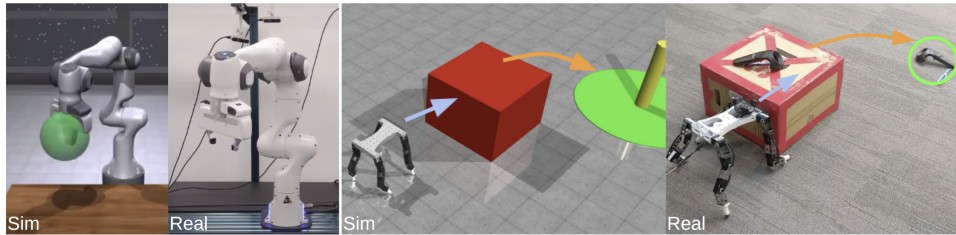

Figure 6: Rigorous measures are taken to capture physical realism in RoboHive's simulations models. Thanks to Robohive's robot class policies trained in simulations easily transfer over to the real world. Depicted in the figure are the synchronized deployment of policies trained in simulation to both simulation and the real world for a representative manipulation task with Franka (left) and a locomotion task with D'Kitty [32] (right).

## 4.3 Teleoperation and Dataset Collection

Since a variety of robot learning methods such as offline reinforcement learning, imitation learning, and finetuning with reinforcement learning rely on meaningfully collected datasets, RoboHive provides both pre-collected datasets - which we refer to as *RoboSet* and teleoperation interfaces for users to collect their own data. The key factors here are the size and diversity of the pre-collected datasets and the ease of use of the teleoperation interface that facilitates dataset collection (Demo ↗) both in simulation as well as the real world.

In addition to refreshed versions of prior datasets ↗ corresponding to the environments we adopted, *RoboSet* also features one of the largest (still growing) open-source real-world robotics datasets ↗ released with commodity hardware covering over 12 skills across over 30 tasks in multiple kitchen scenes. We mark the full composition of *RoboSet* in Table 2. We provide more details on *RoboSet* in Appendix A, the data collection process in subsection A.1, reproducibility statement in subsection A.2, and access and maintenance commitments in subsection A.3.

Table 2: *RoboSet* data compositions across various domains and sources

| Domain | # Trajs | # Tasks | World | Visuals | Source |
|---|---|---|---|---|---|
| Real kitchen | 28,700 | 40 | Real | 4 cam | Human TeleOp [36] |
| Bin Manipulation | 70,000 | 4 | Real | 4 cam | Script+Policy [54] |
| Franka kitchen | 600 | 4 | Sim | 4 cam | Human TeleOp [31] |
| Adroit | 25 | 4 | Sim | 3 cam | Human TeleOp [30] |
| Franka kitchen | 75 | 5 | Sim | 4 cam | Expert Policy [55] |
| Adroit | 75 | 4 | Sim | 3 cam | Expert Policy [55] |
| D'Kitty | 75 | 3 | Sim | 4 cam | Expert Policy [55] |

## 4.4 Datasets and Environments for Visual Imitation Learning and Offline RL

As described above, the ability to directly interact with the tasks has facilitated the collection of a rich dataset in both simulation and the real world. With this dataset, we then benchmark a variety of visual imitation learning (summarized below) and offline RL techniques (see [56] for details).

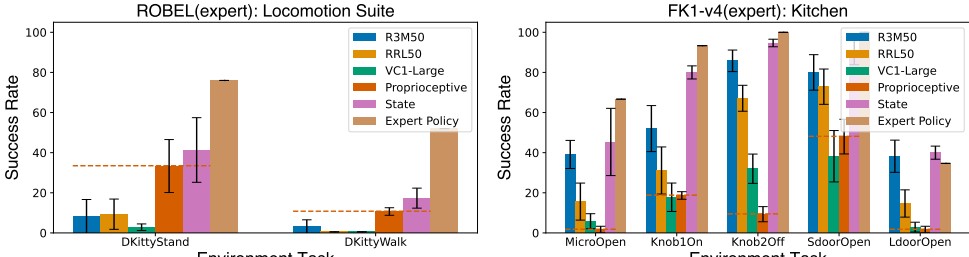

Figure 7: Robel and Kitchen task family: Expert trajectories trained using NPG are used to collect the dataset for behavior cloning consisting of 75 trajectories per task. Each visual baseline in the Kitchen task family is averaged over 3 seeds × 3 camera angles × 25 trajectory rollouts. For the Robel Task family the results are averaged over 3 seeds × 25 trajectory rollouts.

**Learning from Visual Observations** Learning behaviors directly from sensory inputs, especially visual, is challenging due to the high-dimensional, noisy sensory space. Various approaches have been

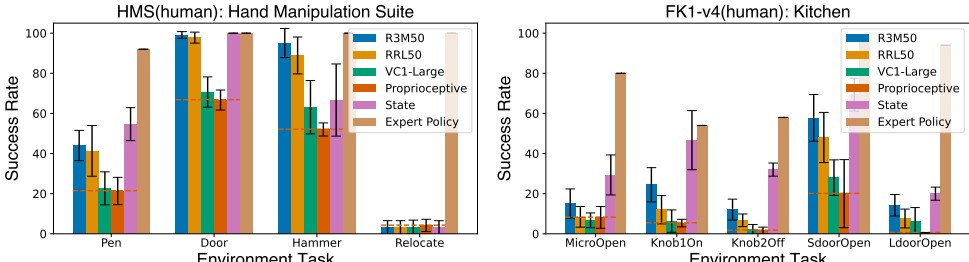

Figure 8: Hand Manipulation and Kitchen task family: Human teleoperation is used to collect the dataset for behavior cloning consisting of 25 trajectories per task for HMS [30] whereas for kitchen tasks, we collect 100 trajectories for each task in Kitchen Task Family (FK1-v4). Each visual baseline in both task families is averaged over 3 seeds × 3 camera angles × 25 trajectory rollouts for robust evaluations.

proposed that learn low-dimensional, compact representations of the input images and then use these representations to learn the task. We benchmark the performance of various state-of-the-art visual representation learning algorithms including RRL [46], R3M [48], and VC1 [50] on the challenging Kitchen domain of Multi-Task, Locomotion Tasks on D'Kitty from ROBEL Task suite, and Adroit domain of the Hand Manipulation [Figure 7, Figure 8] task suites from `RoboHive` using the dataset outlined in Table 2. We compare these visual baselines with an oracle policy trained using privileged 'State' information. Moreover, we add another baseline that is blind to the image observations and uses the robot's proprioceptive information to solve the task, providing a minimum acceptable performance of the visual baselines on the benchmark. While expert demonstrations via an 'Expert Policy' are feasible in simulation and easier to test models, accessing such Expert Policy in the real world is difficult. Alternatively, human-collected demonstrations possessed with multimodal action distribution via teleoperation is widely used in the real world. We benchmark the visual baselines on both types of datasets: Expert Demonstrations (Figure 7) and Human Teleoperated Dataset (Figure 8). The performance of the demonstration trajectories is shown with the label 'Expert Policy,' which consists of rollouts from expert policy in Figure 7 and human teleoperated trajectories in Figure 8. To set the right precedent in the community for a fair and robust comparison of different baselines, we average each method across multiple seeds, and camera angles for all our results.

We observe a wide gap between state-of-the-art visual baselines and policy learned using the privileged state information ('State') [Pink] using both expert and human-teleoperated datasets, highlighting the need for learning better and more robust sensory-motor representations, including visual observations. Surprisingly, the tasks in the locomotion domain (ROBEL in with D'Kitty (Figure 7) perform worse using any visual representation baselines than a proprioceptive-only baseline [Red], highlighting the challenge of learning with diverse visual observations. Moreover, the larger performance gap between tasks in the Kitchen suite using human teleoperated (Figure 8) and dataset collected using expert policy (Figure 8) demonstrates the additional challenges in learning policies that can model multimodal action distributions. Finally, `RoboHive` also includes dexterous Hand Manipulation Suite (Figure 8) where solving tasks with few demonstrations (25 demonstrations in this suite for each task) using behavior cloning can be challenging (e.g., Pen and Relocate). These results underpin the breadth of visual diversity and task-level generalization in `RoboHive`'s task collections often overlooked in prior works.

## 4.5 Well packaged and maintained ecosystem

`RoboHive` offers a versatile simulation and execution tool that caters to various tasks and domains by consolidating all environments under a common base class. The consistent constructor and method signatures make it a one-stop solution. The library's ability to seamlessly integrate environments of diverse natures through a single entry point highlights its versatility.

The library is equipped with a continuous integration workflow that tests each environment construction after each contribution, which ensures that the package is fully functional at any point in time. `RoboHive` is also well packaged and can be instantiated using a `pip install robohive`.

Because a library is only as good as its documentation, `RoboHive` package features a curated and extensive documentation of its features that spans across multiple media. It offers an installation guide☐, a getting started guide☐, a getting started video with simulation as well as real tele-operation demo☐, comprehensive tutorials☐, detailed documentation☐, environment descriptions☐, a video

showcase⧉ of task performance, trained baselines⧉ for different task families, issue tracking⧉ , discussion forums⧉, and support commitments. Depth of `RoboHive` capabilities can be best appreciated by visiting our webpage page - https://sites.google.com/view/robohive⧉.

At the time of writing, the library has been actively maintained for about five years, is mature enough to be used by a broad community of researchers, and has already facilitated a diverse set of results. Given its long-lasting maintenance, we expect `RoboHive`to be actively maintained by its authors as well as the open-source community for the foreseeable future.

### 4.6 Reproducibility

`RoboHive`'s baselines are and will keep on being tracked across various versions of the software to check for performance regressions. In each `RoboHive`'s releases we provide a detailed releases notes⧉ with individual runs and performance comparison across versions with at least 3 seeds for a selected set of environments and algorithms.

## 5 Limitations

With a number of recent publications ([48, 49, 57, 50, 58]) leveraging `RoboHive`, the framework is witnessing a steady adoption in the community. Owing to the scope of the framework, we do have a few limitations. (1) `RoboHive` provide trained agents only for a few algorithmic families that are most commonly used in the field in that corresponding task family. We hope that open-sourcing and community engagements will help us make the coverage more comprehensive. (2) While `RoboHive` has a wide coverage of task families, we hope that the community will help enrich the ecosystem over time, especially towards aerial and navigation suites. (3) `RoboHive`'s robot class supports a diverse set of commonly used hardware but it's not representative. Manipulation hardware support outweighs locomotion, navigation, and aerial hardware support.

## Acknowledgments

Given the broad scope of `RoboHive` and its design flexibility that makes it compatible with a diverse family of algorithms, the true potential of the framework can only be realized with community-wide participation. Hence we are open-sourcing the entire framework. We hope the `RoboHive` ecosystem provides a firm footing for those who are getting started with the field as well as those who are at the forefront of research and development. We sincerely acknowledge the help and support of all the authors and contributors of the original work upon which `RoboHive` builds. `RoboHive` wouldn't have been possible without their original contributions, and their help in interfacing their original work to the `RoboHive` ecosystem.

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

# Appendix

## A Datasets

`RoboHive` offers extensive support for a diverse range of datasets known as *RoboSet*. These datasets are designed to facilitate pre-training and offline learning research. With the support of multiple input devices that users can interact with `RoboHive`'s environments, we obtain datasets that encompass both human-collected data and expert datasets acquired via trained policies.

The dataset is structured in the form of trajectories, capturing essential information at each time step. These trajectories comprise observations, actions, rewards, RGB visuals from multiple camera views, and other relevant environmental information. The richness of information within *RoboSet* makes it suitable for research endeavors across various domains and topics, including but not limited to offline learning like imitation learning and offline reinforcement learning, visual generalization, and policy learning generalization. All datasets within *RoboSet* are in HDF5 format, which is suitable for organizing large and complex hierarchical data.

In the subsequent sections, we introduce the composition of *RoboSet*, the data collection process, and provide insights into our data access and maintenance plan.

### A.1 Collection Process

In this section, we introduce the data collection process of our 1) expert dataset and 2) human dataset. For expert datasets, we use a trained task-specific NPG policy for the target task and roll out 25 trajectories in the environment each for three different trained agents. The expert dataset also contains failure trajectories.

Our human dataset is collected through human teleoperation using Puppet [36]. During the collection process, a human teleoperator uses an HTC Vive headset and controller to control the robot in an end effector space. We subsequently replay and parse the trajectories in each target environment to collect task-relevant information. The human trajectories in *RoboSet* are mostly successful.

Since some of the `RoboHive` environments are built upon existing environments such Adroit and Franka Kitchen, it is natural to integrate datasets collected from the original environments. For instance, our human dataset of the hand manipulation suite is adapted from the human trajectories collected from the DAPG project [51]. We replay the original trajectories into `RoboHive`'s corresponding environments. This enables us to reuse datasets from prior work effectively while supporting information that wasn't contained in the original dataset, for example, RGB observations. By integrating this exteroception information alongside the state-based proprioception observations, `RoboHive` becomes a comprehensive testbed for conducting research on visual generalization as well. To ensure the quality and validity of the replayed trajectories, we provide a reproducibility report in subsection A.2.

### A.2 Reproducibility

The human dataset used in `RoboHive`'s hand manipulation suite is derived from the original dataset from [51]. In order to incorporate this dataset into `RoboHive`, we replay the robot's actions from the original trajectories within the corresponding environment of `RoboHive`. During this process, we record the observations, rewards, and other task-relevant information.

To ensure the validity and reproducibility of the replayed trajectories, we perform a thorough comparison between the 100 original trajectories and the replayed trajectories. Figure 9 depicts the histogram of the norm of the difference between the last states of each original trajectory and the replayed trajectory. Notably, we observe that the trajectories converge to the same environment state within approximation bounds after replay, with the final states being visually indistinguishable. This alignment in performance indicates that the replayed trajectories maintain the same level of quality as the original dataset. As a result, the validity of past results and findings remains intact within the `RoboHive` version of the task. Please refer to our Wiki Page for further information and visualizations.

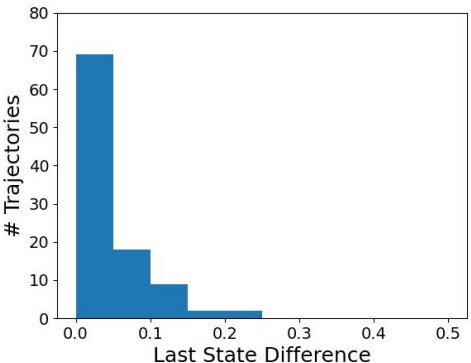

Figure 9: Comparison of Last State Discrepancy between Original and Replay Trajectories

### A.3 Access & Maintenance

To ensure convenient and efficient access for users, we have open-sourced *RoboSet* https://sites.google.com/view/robohive/RoboSet. The complete list of datasets and example trajectories are available on our Wiki Page.

To utilize *RoboSet* effectively, users are expected to download the dataset of their task of interest, and then they can employ their preferred training method to train policies using this data. To evaluate the performance of these trained policies, users are encouraged to perform 25 rollouts in the environment. Lastly, users can compare their methods to our baseline experiments outlined in subsection 4.4.

The availability of both expert and human datasets in *RoboSet* makes it suitable for various learning approaches, encompassing not only imitation learning methods that rely on high-quality data, but also offline reinforcement learning methods and any techniques that leverage multimodal or play data. Moreover, the wealth of visual data within *RoboSet* opens up avenues for visual pre-training and generalization. Importantly, all datasets in *RoboSet* adhere to a consistent format, mitigating the overhead associated with transitioning between different task suites and environments. Looking ahead, we anticipate that *RoboSet* will continue to expand in terms of both quantity and diversity, ensuring its ongoing relevance and value to the research community.

## B Environment sampling performeance

Table 3 illustrates the environment data collection performance on a single machine using 32 processes distributed across 8 A100 GPUs. The data collection process utilized TorchRL's collector classes in an asynchronous manner, following a "first ready first served" approach. Each batch of data (500 steps/batch, 800K total) was collected on a separate process, employing a random policy, and transformed to a floating point tensor and resized to $[84 \times 84]$ pixels. The results presented in this figure are based on three different seeds, and the reported values indicate the average and standard deviation across seeds. Multi-task suite rendering was achieved over 4 cameras with a resolution of $[256 \times 256]$ pixels, Arm and Hand suites had 3 cameras with a resolution of $[244 \times 244]$ pixels. The Myo suite does not support rendering over dedicated cameras and was excluded from this analysis. The decision to employ TorchRL as the backend for data collection is justified by the substantial superiority of these results compared to those achieved using single-process or multiprocessed solutions offered by most other frameworks, as reported in more details in [8].

Table 3: Full details of collection speed for all environments considered on a per-environment basis

| Task Family | Env name | Samples or Frames per second |
|---|---|---|
| Arm Manipulation | *FrankaPickPlaceFixed-v0* | 6174.0235 |
| | *FrankaPickPlaceRandom_v2d-v0* | 928.9412 |
| | *FrankaPickPlaceRandom-v0* | 6127.2433 |
| | *FrankaPushFixed-v0* | 7338.9487 |
| | *FrankaPushRandom_v2d-v0* | 994.2035 |
| | *FrankaPushRandom-v0* | 7382.3643 |
| | *FrankaReachFixed-v0* | 8906.9175 |
| | *FrankaReachRandom_v2d-v0* | 1055.2824 |
| | *FrankaReachRandom-v0* | 8911.4261 |
| Hand manipulation | *door_v2d-v1* | 1681.4599 |
| | *doorv1* | 9124.0997 |
| | *hammer_v2d-v1* | 1722.2912 |
| | *hammer-v1* | 7826.282 |
| | *pen_v2d-v1* | 1711.8543 |
| | *pen-v1* | 8871.5027 |
| | *relocate_v2d-v1* | 1737.4508 |
| | *relocate-v1* | 9317.9963 |
| Multi-task | *FK1_Knob1OnRandom_v2d-v4* | 346.0061 |
| | *FK1_Knob1OnRandom-v4* | 4436.5437 |
| | *FK1_Knob2OffRandom_v2d-v4* | 346.8642 |
| | *FK1_Knob2OffRandom-v4* | 4434.0908 |
| | *FK1_LdoorOpenRandom_v2d-v4* | 346.5334 |
| | *FK1_MicroOpenRandom-v4* | 4424.7308 |
| | *FK1_SdoorOpenRandom_v2d-v4* | 346.0869 |
| | *FK1_SdoorOpenRandom-v4* | 4423.86 |
| Myo | *myoElbowPose1D6MExoRandom-v0* | 11022.1737 |
| | *myoElbowPose1D6MRandom-v0* | 10998.8446 |
| | *myoFingerReachRandom-v0* | 10959.5896 |
| | *myoHandDieReorientRandom-v0* | 6617.4297 |
| | *myoHandKeyTurnRandom-v0* | 7118.316 |
| | *myoHandPenTwirlRandom-v0* | 7890.5196 |
| | *myoHandReachRandom-v0* | 6915.9179 |

