# OpenReview forum: "RoboHive: A Unified Framework for Robot Learning"
_NeurIPS.cc/2023/Track/Datasets_and_Benchmarks — NeurIPS 2023 Datasets and Benchmarks Poster_

### Official Review · Reviewer_NtXK · 2023-07-20
**Review of RoboHive**

**Rating:** 6
**Confidence:** 4
**Correctness:** There are all necessary parts included

**Strengths:**

The RoboHive is large set of tools for development, training and evaluation of wide variety of robotic tasks. it offers separate suites for each of the subfields. The teleoperation methods covers most frequent methods to collect expert trajectories.There is also wide variety of objects and environments. The author put a lot of effort to convert various expert trajectories into one format. I also like the modularity of the system.

**Additional Feedback:**

I reccommand author to add more technical detail in the paper, namely:

1. Description of the simulation engine, physical engine and their limitations (e.g. graspability, contact points)
2. Description of RL methods and details about the training process
3. The brief description how to customise environments and tasks.

**Clarity:**

The paper is well written but in very abstract way and I lack technical details. There are also some repetitive parts in main text and supplement

**Documentation:**

The tool has documentation.

**Ethics:**

There are not any ethic issues as the tool deals with robotic simulation

**Limitations:**

The Limitation section is well described

**Opportunities For Improvement:**

The paper is written more as a PR message than scientific paper. It describes all the advantages of the tool but there are no technical details about implementation presented. I lack detail information about the methods of training, the interface to ML methods and some baselines to the tasks.
The tool offers great collection of robotic taks, on the other hand it is not novel. There is not any extension to a novel tasks or some novel methods to measure the quality of learning.
I laso lack the Information about extendability to a more complex tasks especially  in the multi task suite.

If I compare this tool with others it does not offer any advantage (speed, evaluation, learning methods, extendability, task customisation)

**Relation To Prior Work:**

The tool stems from previous and integrates them together

**Summary And Contributions:**

The paper present benchmark tool for wide variety of robotic tasks, robots and scenarios. The tool is suitable especially for imitation learning as there is many expert trajectories presented in the dataset. The suite offers very good integration of several tools and datasets focusing on collection of expert trajectories from teleoperation. There can be adopted many robots and tasks based on abstract classes for customisation. There is also sim2real interface provided. This tools covers most of the recent tasks in robotic manipulation.

---

> ### Author Response · Authors · 2023-08-23
> **Clarification on contributions and paper format**
>
> We are glad that the reviewers recognized the wide variety of robotic tasks, robots, and scenarios covered in RoboHive. Its impact potential, especially for various learning paradigms, (such as Imitation as well as reinforcement learning), and the associated dataset was also well appreciated. Its features wrt to abstract classes for task customization, modularity, sim2real interface, teleoperation, and unified data format were also well accepted. Additionally, we are excited to hear that the reviewer agrees that RoboHive covers most of the recent tasks in robotic manipulation, although we humbly acknowledge that there is no upper limit and we will keep building forward.
>
> ## Clarification on the paper’s written format
> Given the strong recognition of its strength, the primary concerns from the reviewer seem to be wrt to the written format of the paper. We believe that the confusion stems from strong priors on the format of conference papers. We would like to point to the reviewer that our work is a submission under dataset and benchmarks tasks which clearly states in the [submission instructions](https://neurips.cc/Conferences/2022/CallForDatasetsBenchmarks) that submissions should directly link to repositories/online-material to demonstrate the strength of the submissions, readiness of the proposed work, and to provide a flavor of its usage.
>
> With that in mind, below we highlight a brief list of information already contained in the paper. Notably,
>  Robohive features a curated and extensive documentation of its features that spans across multiple mediums. It offers an
> [installation guide](https://github.com/vikashplus/robohive/tree/main/setup),
> [getting started guide](https://github.com/vikashplus/robohive/tree/main/robohive/tutorials),
> [getting started video with simulation as well as real tele-operation](https://youtu.be/7K03boNPvTM),
> [comprehensive list of tutorials](https://github.com/vikashplus/robohive/wiki/6.-Tutorials-&-FAQs),
> [detailed documentation](https://github.com/vikashplus/robohive/wiki),
> [environment descriptions](https://github.com/vikashplus/robohive/blob/main/robohive/envs/multi_task/README.md),
> [video showcase of results](https://sites.google.com/view/robohive/gallery),
> [pre-trained baselines for different task families](https://github.com/vikashplus/robohive/releases),
> [issue tracking](https://github.com/vikashplus/robohive/issues},
> [discussion forums](https://github.com/vikashplus/robohive/discussions)
>
> The depth of RoboHive capabilities can be best appreciated by visiting our webpage page - [https://sites.google.com/view/robohive](https://sites.google.com/view/robohive\extweblink).
>
>
> ## Framework Extendibility, and task customization
> Framework extendibility and customizations were primary features of RoboHive’s framework. We thank the reviewer for providing us the opportunity to highlight this feature of RoboHive. As we outline in Section-3, RoboHive is extremely modular, and individual components can also be used in isolation. The modularity also facilitates extendibility. The base classes are quite general which allows the extension of new modules. For example, owing to the robot_base class we were able to bring in support for multiple hardware. Owing to the env_base class we were able to bring in support for multiple task suites. Owing to the shared simHive simulations assets, we were able to build comprehensive simulation scenes. RoboHive’s envs follow OpenAI-Gym API – one of the most commonly used frameworks in ML community. All envs are fully customizable ([see Tutorial](https://github.com/vikashplus/robohive/blob/main/robohive/tutorials/1_env_registration_and_customization.ipynb)). Customization are as easy as passing arguments during the registration. We also expose all low-level datastructures to the users that can be edited live during the training to support advanced usecases.
>
> [continued]

---

> > ### Author Response · Authors · 2023-08-23
> >
> > ## Clarification on RoboHive’s contributions
> > We would respectfully like to emphasize that our contributions go much beyond just simple integration and reimplementation of some existing environments. While we are motivated by the choice of well-proven domains, our offerings go much beyond the original work. Using FrankaKitchen introduced by Gupta et al.[1] as an example.
> > Tasks: [1] introduces a single play env. RoboHive offers a multi-task env suite consisting of a diverse collection of single-step as well as multi-step tasks.
> > Scene: [1] introduces scene. Robohive offers multiple choices of scenes from simple tabletop to multiple kitchen
> > Modalities: [1] offers single-state observations. RoboHive offers state, proprioception as well as exteroceptive modalities
> > Dataset: RoboSet took extensive care to maintain the validity of the original dataset from [1]. (This was notoriously difficult to ensure). In addition to the previous dataset being packaged in a unified format, RoboHive also offers collected new datasets for these environments and presents them as part of RoboSet.
> > [Baselines] We present pre-trained baselines for these environments with each release.
> >
> > The fundamental motivation behind Robohive is to facilitate studies towards generalization in robot learning. Towards this end
> > It provides significantly improved physics fidelity as well as visual fidelity.
> > New scenes and environments that weren’t present and introduced in the original works nicely complement the specific vertical to study generalization. We introduce innovative tasks in both simulated and real world environments.  Please refer to our wiki page and webpage for a detailed breakdown of tasks and environments.
> > We present RoboSet one of the largest open-source robotics datasets with multiple skills, tasks, and scenes.
> >
> > For our real-world datasets, we collected 31000 trajectories over a diverse and challenging set of environments. The dataset has been collected from multiple sources, such as kinesthetic demonstrations and teleoperation. We also collected the dataset over multiple camera views, this way we ensure variety in the data and are not bound to a fixed camera viewpoint. For our dataset in simulated environments, we provide both expert trajectories which are collected from task-specific NPG policies as well as human datasets which are collected via human teleoperation. The diverse range of datasets are designed to facilitate pre-training and offline learning research for robotics. Additionally, RoboSet was created to fill the gap that there are no large-scale robotics datasets in a uniform format that is suitable for robot learning.
> >
> > Furthermore, although a subset of RoboHive’s environments are integrated from previous work, we upgrade all of the environments by providing support for visual observations while the original environments only provide state-based observations. This feature of RoboHive opens up avenues for visual pre-training and generalization. For all the datasets within RoboSet, visual observations are included in addition to state-based observations.
> >
> > Last but not least, for all types of robots introduced in RoboHive, we provide hardware support for each of them. This feature is rarely seen in previous environments and benchmarks as they either focus only on simulation or only on real robots. With this support, policies trained in simulations could easily transfer over to the real world, which significantly reduces overhead for researchers who work on both simulated and real robots.
> >
> > Given the diversity of requirements a robot learning stack needs, we hope the diversity of offering a simple “pip insall robohive” command provides convinces the review of multiple boundaries RoboHive will transcend towards unifying and standardizing robot learning.
> >
> > [continued]

---

> > > ### Author Response · Authors · 2023-08-23
> > >
> > > ## Response on specific asks
> > > Based on the reviewer’s feedback, we are improving our writing adding technical details and specifics *throughout* the paper. Below we answer their specific questions
> > > 1. Description of simulation engines and their limitations.
> > > The details of our simulation engine are outlined in sec 3.1. Specifically, our simulations are powered by MuJoCo – one of the most well-used physics engines in the Robot Learning community. Since we directly use the official releases, we directly borrow all its features. A comparison of physics engines is well beyond the scope of our submissions, we point the reviewer to [MuJoCo’s official documentation](https://mujoco.org) for full details of MuJoCo, and the following papers [1][2][3] for an unbiased comparison of different physics engines.
> > >
> > > 2. Description of RL methods and details about the training process
> > > RoboHive is a comprehensive framework with multiple usecases well beyond just reinforcement learning. We provide an exhaustive list of common usecases in Section 4 in the paper. Full details on all our baselines (including RL) can be found at [RoboHive agent’s page](https://sites.google.com/view/robohive/baseline) which contains details on both – [how to use and take benefit of our pretrained agents](https://sites.google.com/view/robohive/baseline?authuser=0#h.uj2k6wef2ex7), as well as [how to train your own agent](https://sites.google.com/view/robohive/baseline?authuser=0#h.1znptj4y557q). RoboHive environments are simple OpenAI-Gym environments and are fully compatible with any framework that supports Gym-API. We also provide a [getting started COLAB](https://www.google.com/url?q=https://colab.research.google.com/drive/1rdSgnsfUaE-eFLjAkFHeqfUWzAK8ruTs?usp%3Dsharing&sa=D&source=docs&ust=1692802691158780&usg=AOvVaw2DD5DmCIgR7f7uHMYYIDFH) that allows uses to quickly get started on our environments, we also provide [COLAB demonstrating how to train agents online](https://www.google.com/url?q=https://colab.research.google.com/drive/1NHpellroUkpX7eaDVErGbBdqLHxGGhYW?usp%3Dsharing&sa=D&source=docs&ust=1692802691158929&usg=AOvVaw3GyemNd9GkeqDAqDN95D7c) without even needing any local setups! Our agent’s framework [AgentHive](https://github.com/facebookresearch/agenthive) supporting multiple frameworks TorchRL, mjRL, Stable Baselines is fully documented and open-source as well for reproducibility.
> > >
> > > With over an order more environments (RoboHive has ~500 environments) across multiple task suites and the diversity of algorithmic paradigms, it's near impossible to provide exhaustive coverage over its use cases. This is precisely the motivation behind open-sourcing RoboHive to the entire community. We are looking forward to different ways the community will leverage RoboHive.
> > >
> > > 3. A brief description of how to customize environments and tasks.
> > > RoboHive hosts a [large collection of Tutorials covering multiple use cases](https://github.com/vikashplus/robohive/wiki/6.-Tutorials-&-FAQs). We have an explicit [tutorial for env registration and customization](https://github.com/vikashplus/robohive/blob/main/robohive/tutorials/1_env_registration_and_customization.ipynb) (also linked in Section 4.5 of main paper).
> > >
> > >
> > > [1] Todorov et al. Mujoco: A physics engine for model-based control 2012
> > > [2] Erez et al. Simulation tools for model-based robotics: Comparison of bullet, havok, mujoco, ode and physx 2015
> > > [3] Chung et al. Predictable behavior during contact simulation: a comparison of selected physics engines 2016
> > >
> > >
> > > Given the breadth of RoboHive constraints of the page limit, we have deliberated hard on a balanced way to wisely distribute details across the paper and link to documentation where appropriate. We hope our response mitigates reviewers’ concerns. We welcome additional inputs and concrete suggestions to improve organization and are eager to find a way to converge on a presentation format that will best position RoboHive to be widely adopted and useful for the Robot Learning community.

---

> > > > ### Author Response · Authors · 2023-08-28
> > > > **Review reminder**
> > > >
> > > > Dear Reviewer,
> > > >
> > > > We wish to place a gentle reminder that we have submitted our rebuttal in response to the feedback and suggestions you provided. It would be highly valuable for us if you could find some time for a review. We have specifically addressed the critical points around the writing format we adopted in the paper and pointers to all relevant requested details. Please let us know whether we have adequately addressed your concerns.
> > > >
> > > > We genuinely appreciate the time and effort you've put into reviewing our work. If you find our responses satisfactorily address your concerns and feedback, we kindly invite you to consider adjusting the score accordingly.
> > > >
> > > > Best regards, The Authors

---

> > > > ### Comment · Reviewer_NtXK · 2023-08-29
> > > > **Reply to the authors**
> > > >
> > > > Dear authors,
> > > >
> > > > thanks for the clarification. I really appretiate the effort you put into this toolbox.
> > > >
> > > > "Description of RL methods and details about the training process RoboHive is a comprehensive framework with multiple usecases well beyond just reinforcement learning. We provide an exhaustive list of common usecases in Section 4 in the paper. Full details on all our baselines (including RL) can be found at RoboHive agent’s page which contains details on both – how to use and take benefit of our pretrained agents, as well as how to train your own agent. RoboHive environments are simple OpenAI-Gym environments and are fully compatible with any framework that supports Gym-API. We also provide a getting started COLAB that allows uses to quickly get started on our environments, we also provide COLAB demonstrating how to train agents online without even needing any local setups! Our agent’s framework AgentHive supporting multiple frameworks TorchRL, mjRL, Stable Baselines is fully documented and open-source as well for reproducibility."
> > > >
> > > > I checked the pretrained agents and they are able to learn all simple tasks with 100 success rate or to learn few tasks with 50 percent success rate. The rest of tasks are non-solvable by recent algorithms. The results are similar to many other tools (RLBench,Robosuite, OmniGibson) where simple tasks are perfectly trainable but there is not any relation among tasks. I am missing the results for multi-step suite in the baselines section.
> > > > The main disadvantage of your tool is similar to previous tools. Moreover RoboHive is collection of benchmarks from different unrelated robotic areas (you cannot adopt trained modules from one suite to another). It does not offer scalability to more complex tasks. The user will choose one of the suites (according to his/her research area) and there is random collection of simple or non-solvable tasks.
> > > >  I know it is difficult to create curriculum of tasks with increasing complexity (especially in mulptiple suites), but without this feature there is not any novelty compared to previous ones (e.g. OmiGibson) other than coverage of unconnected areas (Myo, Kitchen, Locomotion, Hand, Multi-step).

---

> > > > > ### Author Response · Authors · 2023-08-29
> > > > >
> > > > > Thank you for your response.
> > > > >
> > > > > For any platform to be a good candidate for research, we believe there has to be a healthy representation of tasks spanning from easy to hard. This suggests that the point raised by the reviewers is a Pro instead of a Con of our framework.
> > > > >
> > > > > Additionally, the characterization provided in response is a case of over-generalization and digresses away from the claims we have made in the paper. For example -
> > > > > 1. "Categorization of all tasks that can be solved as simple as hence not useful":  We would like to emphasize that the contribution of a task is incomplete without the context of the problem/question it's being studied in association with. For instance, tasks that are easy to solve via state or dense rewards can become prohibitively hard to solve via visual inputs or sparse rewards. RoboHive supports all these variations of every task to facilitate such investigations.
> > > > > 2. "RoboHive is a collection of benchmarks from different unrelated robotic areas. Trained modules can't be adopted between modules": RoboHive's claims are not built around transfer as our key feature. While generally hard to do between arbitrary sets, interesting results in recent work have shown compelling transfer capabilities, for example, representation learning methods like R3M, VC1, VIP etc. are showing strong transfer of features between disjoint sets, even human videos. Moreover, our single task suite (arms, etc.) can serve as a bootstrapping mechanism for multi-task, cross-embodiment learning techniques.
> > > > >
> > > > > With regards to concerns about novelty as compared to existing tools, we would like to emphasize that RoboHive is positioned as an integrated platform/ecosystem that can cater to different requirements of the robot learning community. We would like to emphasize several points of difference - 1) While benchmarks such as Omnigibson provide a framework for just mobile manipulation in the home, RoboHive offers a much more diverse collection of environments across domains. This enables researchers in domains like locomotion, dexterous manipulation and tabletop manipulation who use MuJoCo as their choice of simulation to have a unified platform for their robotic learning infrastructure. 2) RoboHive offers an unprecedented level of physical realism, especially for tasks involving significant amounts of contact or dexterity, 3) RoboHive is a tool specifically built for enabling robotic learning work - so it naturally provides teleoperation and hardware hooks, enabling seamless transitions between simulated robots and their physical counterparts. Moreover, the code is trivially easy to plug into any reinforcement learning or imitation learning framework. To our knowledge, no other platform offers such a vast collection of realistic environments and functionality.
> > > > >
> > > > > RoboHive offers a diverse collection of environments, clearly defined task descriptions, and evaluation protocols. This facilitiates algorithmic research in RL, imitation learning, world models etc. while grounded in realistic environments.
> > > > > All the scenes and tasks are carefully tuned not only for visual richness but also for high physics fidelity.
> > > > > Beyond benchmarking, as a research tool, RoboHive provides teleoperation and hardware hooks, enabling seamless transitions betweens simulated robots and their physical counterparts.
> > > > > To our knowledge, no other platform (e.g. IssacSim, Habitat) offers such a vast collection of realistic environments and functionality.
> > > > >
> > > > > In our response above labeled "Framework Extendibility, and Task Customization" we provide details on the task's modularity and customized. Unlike reviews' beliefs, we strongly emphasize that creating such curriculums is as easy as passing arguments during environmental registration. For example
> > > > > 1. Initialization and goal ranges are fully customizable: To create a curriculum users can start from small ranges and gradually widen it. Indeed we use this feature to train various agents towards 100% success (which the reviewer has incorrectly claimed makes the tasks easy).
> > > > > 2. Rewards terms, weights, and modes are fully customizable: To create a curriculum one can start by providing rewards for small achievements and gradually move towards broader goals (e.g.: stand> walk> running).
> > > > > 3. Multi-step environments are fully customizable: The user can start by solving a single step and slowly introduce more steps.
> > > > >
> > > > > We agree with the reviewer that we don't solve *all* the challenges present in robot learning. Nonetheless, RoboHive with its range of offerings from the diversity of environments across multiple suites, environments designed for visual and task generalization, diverse datasets, hardware support, etc. uniquely positions it to make a strong platform for collaborative research in its current form. Given its scope, for RoboHive to reach its true potential, it needs community engagement. We hope the reviewers can provide us suggestions for how we can improve our key claims.

---

### Official Review · Reviewer_iKLG · 2023-07-20
**Review of the paper "RoboHive: A Unified Framework for Robot Learning"**

**Rating:** 7
**Confidence:** 3
**Correctness:** yes
**Clarity:** yes

**Strengths:**

1. Integrate existing robot learning tasks and coding under a unified architecture for researchers to use and reproduce.
2, Provide the interface of robot teleoperation that can collect demonstration, which is not supported by some original environments.
3, Provides higher physics fidelity and rich visual diversity content, improving the performance of sim2real.

**Additional Feedback:**

none

**Documentation:**

yes

**Limitations:**

yes

**Opportunities For Improvement:**

1, Teleoperation support is a good feature. I see that RoboHive has many dexterous hand environments, how does the dexterous hand collect demonstration? is it using the CyberGlove mentioned in the paper? And I don't see the dexterous hand's teleoperation in the short video.
2, As a benchmark paper, I think the tested algorithm and environment in "Learning from Visual Observations" are a bit less. As I understand, the paper just tested the performance of RRL, R3M, and VC1 in the Hand Manipulation and Kitchen task family environment.
3, Although this paper provides a lot of new features, it is still just the integration and reimplementation of some existing environments, and I still think that there is a bit of a lack of novelty.

**Relation To Prior Work:**

yes

**Summary And Contributions:**

This paper presents a unified robot learning framework RoboHive. RoboHive improves and integrates with the existing RL environment, and adds new content, such as a unified robot class that might benefit for sim2real, teleoperation, more realistic physics and visual rendering, and benchmark metrics. RoboHive covers a large variety of robots and their tasks and benchmarks some visual representation learning algorithms such as RRL, R3M, and VC1. It provides a new standard benchmark for algorithms in the field of robot learning.

---

> ### Author Response · Authors · 2023-08-21
> **Response to reviewer iKLG**
>
> Thank you for taking valuable time to review our paper and sharing feedback. We are glad that you find various features of RoboHive to be valuable! Your feedback has greatly helped us in improving the paper writing and highlighting the contributions. We respond to your suggestions below.
>
> **Dexterous hand teleoperation** You are correct, the TeleOp demonstrations are indeed collected using CyberGlove! We value your feedback and will make this detail more clear/prominent in the paper. A short demo of dexterous hand teleoperation can be viewed through [this link](https://youtu.be/mic9l7h-xts) and we will add this to the website.
>
> **Clarification on benchmarking algorithms**
>
> We first clarify that RoboHive is positioned as an integrated platform/ecosystem that can cater to different requirements of the robot learning community. As a benchmark, RoboHive offers a diverse collection of environments, clearly defined task descriptions, and evaluation protocols. This facilitates algorithmic research in RL, imitation learning, world models etc. while grounded in realistic environments. RoboHive-0.6 contains ~500 environments across the different robots, tasks, and scenes. While we agree with you that it will be amazing to have a large collection of algorithms benchmarked, it is practically infeasible for any single research team to comprehensively run a large collection of baselines across all the environments. To date we have done the next best things possible.
> - In section 4 we have given concrete evidence of multiple use cases we support using a subset of environments.
> - With each [release](https://github.com/vikashplus/robohive/releases) we provide an extensive collection of pre-trained baselines which also help us track improvements/regressions over time.
>
> We also note that our ultimate vision with RoboHive is to enable different sub-communities of robot learning to come together, standardize, collaborate, and share results. As RoboHive gets widely adopted, we hope that comprehensive benchmarking across environments and algorithms will emerge with contributions from everyone in the community.
>
> **Clarification on contributions**
>
> We respectfully emphasize that **our contributions go much beyond just simple integration** and reimplementation of existing environments. While our choices are motivated by well-proven domains, our offerings go much beyond the original work. Using FrankaKitchen introduced by Gupta et al. [1] as an example,
> - **Tasks:** [1] introduces a single play env. In contrast, RoboHive offers a multi-task env suite consisting of a diverse collection of single-step as well as multi-step tasks.
> - **Scene:** [1] introduces a single scene vs multiple kitchen scenes and tabletops in RoboHive
> - **Modalities:** [1] offers single-state observations.RoboHive offers state, proprioception as well as exteroceptive modalities
> - **Dataset:** RoboSet took extensive care to maintain the validity of the original dataset from [1]. (This was notoriously difficult to ensure). In addition to the previous dataset being packaged in a unified format, RoboHive also collects and shares a much larger dataset as part of [RoboSet](https://sites.google.com/view/robohive/roboset), which is at least 10x larger in size.
>
> Similar functionality offerings and extensions over prior work are available across the various suites in RoboHive.
>
> We also highlight that various components of a robot learning stack have been notoriously hard to install, often with conflicting dependencies and incompatibilities. Anecdotally, we have observed that several students starting their research journey are dismayed by the difficulty of getting on-boarded to a robotics project or stack. We take great pride in the simplicity of RoboHive’s installation process. All the features can be accessed using a simple `pip install robohive` command, thereby greatly reducing the barrier of entry.
>
> [1] Gupta et al. Relay Policy Learning: Solving Long-Horizon Tasks via Imitation and Reinforcement Learning. CoRL 2019.

---

> > ### Comment · Reviewer_iKLG · 2023-08-22
> > **Official Comment by Reviewer iKLG**
> >
> > Thank you for taking the time to answer my questions and address my comments. I do not have any further questions. Please make sure that all the revisions are included in the final release. This is important not only for the paper's quality but also for the community to better understand all the details.

---

### Official Review · Reviewer_UW1F · 2023-07-20
**a comprehensive framework bridging the gap in robot learning research**

**Rating:** 8
**Confidence:** 3
**Correctness:** Yes.

**Strengths:**

This paper contributes to the robot learning field by introducing a comprehensive and diverse framework that bridges gaps in existing benchmarks, and integrates both physical and simulated robotics. It provides a unified, accessible platform, thus promoting collaboration, standardization, and rapid prototyping across diverse areas of robot learning.

**Additional Feedback:**

please see above.

**Clarity:**

Yes. The figures are nicely rendered and organized, and the charts, such as Figure 1, 3, 4, 7, are all very helpful for the clarity of the paper.

**Documentation:**

Yes. All materials are provided with great clarity and thoroughness.

**Ethics:**

No.

**Limitations:**

The authors did a nice job discussing RoboHive's limitations in its limited offering of trained agents for specific algorithmic families and an uneven representation in task and hardware support.

**Opportunities For Improvement:**

The paper would benefit from additional information on the performance of the simulator across a range of scenarios, as this forms a critical component of the proposed framework. Notably, MuJoCo, with its focus on real-time simulation, may trade off some accuracy for efficiency, which often requires per-scenario parameter tuning to replicate real-world conditions. It would be helpful if the authors could shed some light on the extent of parameter tuning required when utilizing RoboHive and whether the maximum achievable simulator accuracy is sufficiently high for effective sim-to-real tasks.

**Relation To Prior Work:**

Yes, the authors effectively discussed how their work stands out from previous contributions by offering a more comprehensive, diverse, and physically accurate benchmark that encourages broader participation and rapid innovation in robot learning research.

**Summary And Contributions:**

The paper presents RoboHive, a unified and comprehensive framework designed to address the complexities and accessibility issues hindering progress in robot learning. RoboHive functions as both a research tool and a simulation benchmark, providing diverse environments, well-defined tasks, robust evaluation protocols, and supporting various learning paradigms. It seamlessly integrates physical robots and simulated counterparts, enabling the effective deployment and testing of policies in real-world scenarios and simulations.

---

> ### Author Response · Authors · 2023-08-21
> **Thank you for the kind feedback. PFB responses to your question.**
>
> Thank you for taking valuable time to review our paper. We appreciate your recognition of the comprehensiveness of RoboHive and the diversity of environments and capabilities. We resonate with your vision that by providing a unified and accessible platform, RoboHive is well positioned to bridge different sub-communities, promote collaboration, standardization, and rapid prototyping across robot learning.
>
> Simulation as well as simulation fidelity is a important factor in Robotics. RoboHive’s inception was indeed motivated by struggles with lack of well tuned physics in robotics benchmarks. We expect RoboHive’s offering to be an out of box solution requiring little-to-no paramater tuning. We thank the reviews for providing an opportunity to highlight our contributions towards physics fidelity. Given the complexity of environments and the contact-rich nature of benchmark tasks encapsulated in RoboHive, physics fidelity is of utmost important to us. All scenes and tasks are carefully tuned not only for visual richness but also for high physics fidelity.
>
> An additional but noteworthy detail is that RoboHive tasks leverage a shared library of simulation models (referred to as ["simhive"](https://github.com/vikashplus/robohive/tree/main/robohive/simhive)) that cuts across multiple tasks. This implies that various permutations and combinations of robots, scenes, and tasks can be explored to create environments, but the physics need to be tuned only once per component. This strategy has empowered us to subject our simulations to rigorous physics accuracy assessments across diverse environments and numerous projects developed over the course of RoboHive's four-year evolution.
>
> The effectiveness of our simulations is further underscored by a substantial number of sim2real outcomes that we showcase in the [results gallery on our website](https://sites.google.com/view/robohive/gallery/real-gallery).

---

> > ### Author Response · Authors · 2023-08-28
> > **Review reminder**
> >
> > Dear Reviewer,
> >
> > We wish to place a gentle reminder that we have submitted our rebuttal in response to the feedback and suggestions you provided. It would be highly valuable for us if you could find some time for a review. Please let us know whether we have adequately addressed your concerns.
> >
> > We genuinely appreciate the time and effort you've put into reviewing our work. If you find our responses satisfactorily address your concerns and feedback, we kindly invite you to consider adjusting the score accordingly.
> >
> > Best regards, The Authors

---

> > > ### Comment · Reviewer_UW1F · 2023-08-30
> > > **Thank you for your reply**
> > >
> > > Dear Authors,
> > >
> > > Thank you for your comprehensive response to my review. I appreciate the additional information provided. Your clarifications on parameter tuning and the "simhive" library add meaningful depth to the paper and provide a better understanding of the framework's robustness.
> > >
> > > I'm pleased to see that the issues I raised have been carefully considered and thoroughly addressed in your reply. Your focus on physics fidelity and the substantial number of sim2real outcomes further solidify the contributions of RoboHive to the field of robot learning.
> > >
> > > After reviewing your response, I have decided to maintain my original score for the paper. I believe the paper makes a significant contribution to the field, and I look forward to seeing RoboHive's impact in fostering collaboration and innovation across various sub-domains of robot learning.
> > >
> > > Best regards,
> > > Reviewer

---

### Official Review · Reviewer_DRrC · 2023-07-28
**RoboHive review**

**Rating:** 5
**Confidence:** 3
**Correctness:** Seems correct.
**Clarity:** Fairly clear.

**Strengths:**

1. This paper introduces a unified framework for learning tasks in robotics filed.
2. Various tasks have been implemented to demonstrate the effectiveness of this framework.

**Additional Feedback:**

Nothing particular.

**Documentation:**

The link to the website is provided.

**Ethics:**

Not applicable.

**Limitations:**

1. A detailed explanation for sim-to-real needs to be added in the paper. For example, how the framework integrates the delays and dynamic errors between simulation and physical robots.
2. In the visual observations, it is meaningful to compared this framework with previous approach in terms of the simulation speed instead of simply comparing between different tasks inside the framework.
3. For the supplementary material, it would be great if the authors include a detailed documentation to use the material and put them onto the public repo.


**Opportunities For Improvement:**

See limitations.

**Relation To Prior Work:**

The paper discussed and compared with the prior work sufficiently.

**Summary And Contributions:**

This paper proposes a comprehensive pipeline for learning in robotics community. This paper showcases the multiple application scenarios, such as manipulation, locomotion, swarm robotics, and musculoskeletal agents in the simulation environment. The pipeline unifies the hardware and software into this framework, and implements the interface API. R3M, RRL, VC1 and expert policy are compared for hand manipulation tasks and Kitchen tasks.

---

> ### Author Response · Authors · 2023-08-21
> **Response to Reviewer DRrC**
>
> Thank you for your insightful review. We appreciate your recognition of RoboHive's value as a unified robot learning platform and its support for diverse environments. Your input has significantly improved our paper's presentation. We first clarify a few important details below, and then address your specific questions.
>
> **Documentation** Due to the nature of the track/submission, we are constrained to provide "high-level" details within the paper. Detailed information on RoboHive can be accessed through the [project Wiki](https://github.com/vikashplus/robohive/wiki), complemented by an array of videos on the [accompanying website](https://sites.google.com/view/robohive/gallery) which showcase various environments and capabilities. We request the reviewers to explore these resources for a comprehensive assessment of the submission.
>
> **Positioning** RoboHive is positioned as an integrated platform/ecosystem that can cater to different requirements of the robot learning community. As a benchmark, RoboHive offers a diverse collection of environments, clearly defined task descriptions, and evaluation protocols. This facilitiates algorithmic research in RL, imitation learning, world models etc. while grounded in realistic environments. As a research tool, RoboHive provides teleoperation and hardware hooks, enabling seamless transitions betweens simulated robots and their physical counterparts.
>
> With this context, we answer your specific questions below.
>
> **Q1: A detailed explanation for sim-to-real needs to be added in the paper. For example, how the framework integrates the delays and dynamic errors between simulation and physical robots.**
>
> We provide detailed documentation about features that can facilitate sim2real in RoboHive's [wiki](https://github.com/vikashplus/robohive/wiki) and [FAQ section](https://github.com/vikashplus/robohive/wiki/6.-Tutorials-&-FAQs). For instance, RoboHive natively supports visual domain randomization and scene layout randomization in several environments. Based on reviewer feedback, we will add a high-level summary of components that can aid sim2real transfer in the appendix. Some highlights of the components/features supported are:
>
> **Domain Randomization**
> - RoboHive supports all standard modalities for domain randomization including variations in physical properties (e.g. mass, friction, sizes), object visuals (e.g. color, textures) and scene visual properties (e.g. illumination, color tones). Users are also provided low level access to all simulation assets that can be manipulated directly during the runtime to facilitate needs for custom variations.
> - Real-world variations such as delays and sensors noise are also supported. Working with delayed signal is as easy as specifying delay while querying observations `env.get_obs(dealy=-2)`. All input streams also have a noise generator attached to them which can be individually configured using a single config file, can be globally controlled using a single variable (noise scale).
> - All our domain randomization features can be configured ahead of time, or can be changed in real time during execution.
>
> **Sim2Real Bridge**
> - To facilitate a seamless transition between simulation and the real world, we have devised a unified robot class (see Sections 3.1 and 4.2). This class acts as an effortless bridge, enabling swift toggling between simulation and reality. A [demonstrative video](https://sites.google.com/view/robohive/demo) of this process is available on the project's website.
>
> **Q2: regarding simulation speeds**
>
> Distinct from platforms like Habitat and RoboSuite which utilize different components for simulation and visualization, RoboHive directly interfaces with MuJoCo for both simulation and visualization with no extra overhead. Since both prior platforms and RoboHive directly rely on MuJoCo, there will be no difference in simulation speed for any given scene.
>
> **Q3: In the supplementary material, it would be beneficial for the authors to provide comprehensive documentation for utilizing the materials and integrating them into the public repository.**
>
> We wholeheartedly concur with the reviewer's perspective and take pride in the simplicity of our offering (`pip install robohive`), complemented by our comprehensive [public documentation](https://github.com/vikashplus/robohive/wiki). This encompass an extensive documentation wiki, an installation guide, a user-friendly initiation guide complete with teleoperation demonstrations in both simulated and real environments, in-depth tutorials elucidating features, detailed environment descriptions, task performance videos, established baselines for distinct task categories, avenues for issue tracking, discussion forums, and support commitments. Following reviewer suggestions, we have also incorporated Colab tutorials to facilitate direct online experimentation with RoboHive, improving ease-of-access and adoption. We will include pointers to all these resources in the paper.

---

> > ### Author Response · Authors · 2023-08-28
> > **Review reminder**
> >
> > Dear Reviewer,
> >
> > We wish to place a gentle reminder that we have submitted our rebuttal in response to the feedback and suggestions you provided. It would be highly valuable for us if you could find some time for a review. We have specifically addressed the critical points you highlighted, such as sim2real needs and comprehensive documentation. Please let us know whether we have adequately addressed your concerns.
> >
> > We genuinely appreciate the time and effort you've put into reviewing our work. If you find our responses satisfactorily address your concerns and feedback, we kindly invite you to consider adjusting the score accordingly.
> >
> > Best regards,
> > The Authors

---

### Author Response · Authors · 2023-08-29
**Summary of responses to AC and reviewers**

We thank all reviewers and AC for reviewing our paper and sharing feedback. We wanted to summarize some some of the main points from our responses.

**On the positioning of RoboHive:**  We believe some reviewers have misunderstood the positioning of our paper and the core contributions. RoboHive is positioned as an integrated platform/ecosystem that can cater to different requirements of the robot learning community.
- As a benchmark, RoboHive offers a diverse collection of environments, clearly defined task descriptions, and evaluation protocols. This facilitiates algorithmic research in RL, imitation learning, world models etc. while grounded in realistic environments.
- All the scenes and tasks are carefully tuned not only for visual richness but also for high physics fidelity.
- Beyond benchmarking, as a research tool, RoboHive provides teleoperation and hardware hooks, enabling seamless transitions betweens simulated robots and their physical counterparts.
- To our knowledge, no other platform (e.g. IssacSim, Habitat) offers such a vast collection of realistic environments and functionality.

**Our contributions**  We respectfully emphasize that our contributions go much beyond just simple integration and reimplementation of existing environments. While our choices are motivated by well-proven domains, our offerings go much beyond the original work. Using FrankaKitchen introduced by Gupta et al. [1] as an example,
- **Tasks:** [1] introduces a single play env. In contrast, RoboHive offers a multi-task env suite consisting of a diverse collection of single-step as well as multi-step tasks.
- **Scene:** [1] introduces a single scene vs multiple kitchen scenes and tabletops in RoboHive
- **Modalities:** [1] offers single-state observations. RoboHive offers state, proprioception as well as exteroceptive modalities
- **Dataset:** We took extreme care to preserve validity of data from [1]. In addition to the previous dataset being packaged in a unified format, RoboHive also collects and shares a much larger dataset as part of [RoboSet](https://sites.google.com/view/robohive/roboset), which is at least 10x larger in size.

**On Documentation**  Due to the nature of the track/submission, we are constrained to provide "high-level" details within the paper. Detailed information on RoboHive can be accessed through the [project Wiki](https://github.com/vikashplus/robohive/wiki), complemented by an array of videos on the accompanying [website](https://sites.google.com/view/robohive) which showcase various environments and capabilities. We request the reviewers to explore these resources for a comprehensive assessment of the submission.

**Ease of Installation**  We also highlight that various components of a robot learning stack have been notoriously hard to install, often with conflicting dependencies and incompatibilities. Anecdotally, we have observed that several students starting their research journey are dismayed by the difficulty of getting on-boarded to a robotics project or stack. We take great pride in the simplicity of RoboHive’s installation process. All the features can be accessed using a simple `pip install robohive` command, thereby greatly reducing the barrier of entry.

[1] Gupta et al. Relay Policy Learning: Solving Long-Horizon Tasks via Imitation and Reinforcement Learning. CoRL 2019.

---

### Decision · Program_Chairs · 2023-09-22

**Decision:**

Accept (Poster)

**Comment:**

Broadly speaking, this paper presents a potentially useful framework for the exploration of simulated tasks in the robotics arena. It integrates a number of distinct tasks with a well-known physics simulator and provides a comparatively simple entry point to starting work on sim-to-real robotics efforts. Although this work is similar to other efforts to build robotic simulation environments, the suite of tasks and efforts to ensure physical fidelity are noteworthy. However, the scientific novelty of the benchmark itself is somewhat limited. There are also areas in which the paper itself could be improved, especially with the addition of more technical detail; the presentation of the paper in a benchmarks track does not in and of itself mean that technical content is inappropriate. More discussion of the limitations of the simulator, especially limitations that are inherent to MuJoCo, would be informative. The authors are encouraged to take reviewers' detailed suggestions into account when preparing future versions of the paper.

Pros:
+ The framework for learning tasks in robotics is in an important area, and RoboHive supports a wide range of tasks that could in turn support sim-to-real development in robotics.
+ The (claimed) simple to begin working with mechanism for robot-based learning has the potential to improve accessibility for this kind of research.
+ Allows for the collection of demonstrations of tasks, which is a difficult problem hat some competitor technologies do not support.
+ A wide variety of objects and environments is represented.

Cons:
- Technical and implementation content could be more detailed. Papers in the benchmarks track present things at a high level, yes, but it is still a technical track.
- Limitations of the toolkit should be made clearer, especially wrt. simulation speed and the accuracy of the underlying simulator.
- Some discussion of the parameter tuning required when using the toolkit would be informative.
- RoboHive offers a significant possible set of tasks, but is not in and of itself scientifically novel. More demonstration of effective sim-to-real performance would help clarify the contribution.